# Unveiling the Role of PAR 1: A Crucial Link with Inflammation in Diabetic Subjects with COVID-19

**DOI:** 10.3390/ph17040454

**Published:** 2024-04-01

**Authors:** Ravinder Singh, Varinder Singh, Md. Altamash Ahmad, Chirag Pasricha, Pratima Kumari, Thakur Gurjeet Singh, Rupinder Kaur, Somdutt Mujwar, Tanveer A. Wani, Seema Zargar

**Affiliations:** 1Chitkara College of Pharmacy, Chitkara University, Rajpura 140401, Punjab, India; singhvarinder28nov@gmail.com (V.S.); altamash18001.ccp@chitkara.edu.in (M.A.A.); chiragpasricha10825@gmail.com (C.P.); pratima.kumari@chitkara.edu.in (P.K.); gurjeet.singh@chitkara.edu.in (T.G.S.); rupinder@chitkara.edu.in (R.K.); somduttmujwar@gmail.com (S.M.); 2Department of Pharmaceutical Chemistry, College of Pharmacy, King Saud University, Riyadh 11451, Saudi Arabia; twani@ksu.edu.sa; 3Department of Biochemistry, College of Science, King Saud University, Riyadh 11495, Saudi Arabia; szargar@ksu.edu.sa

**Keywords:** protease-activated receptor 1, COVID-19, inflammation, diabetes mellitus, cytokines

## Abstract

Inflammation is a distinguished clinical manifestation of COVID-19 and type 2 diabetes mellitus (T2DM), often associated with inflammatory dysfunctions, insulin resistance, metabolic dysregulation, and other complications. The present study aims to test the hypothesis that serum concentrations of PAR-1 levels differ between COVID-19 diabetic patients (T2DM) and non-diabetic COVID-19 patients and determine their association with different biochemical parameters and inflammatory biomarkers. T2DM patients with COVID-19 (n = 50) with glycated hemoglobin (HbA1c) levels of (9.23 ± 1.66) and non-diabetic COVID-19 patients (n = 50) with HbA1c levels (4.39 ± 0.57) were recruited in this study. The serum PAR-1 levels (ELISA method) were determined in both groups and correlated with parameters such as age, BMI, inflammatory markers including CRP, interleukin 6 (IL-6), tumor necrosis factor-alpha (TNF-α), D-dimer, homocysteine, and N-terminal pro–B-type natriuretic peptide (NT-proBNP). Demographic variables such as BMI (29.21 ± 3.52 vs. controls 21.30 ± 2.11) and HbA1c (9.23 ± 1.66 vs. controls 4.39 ± 0.57) were found to be statistically elevated in COVID-19 T2DM patients compared to non-diabetic COVID-19 patients. The concentrations of several inflammatory biomarkers and PAR-1 were remarkably increased in the COVID-19 T2DM group when compared with the non-diabetic COVID-19 group. The univariate analysis revealed that increased serum PAR-1 estimations were positively correlated with enhanced HbA1c, BMI, inflammatory cytokines, D-dimer, homocysteine, and NT-proBNP. The findings in the current study suggest that increased levels of serum PAR-1 in the bloodstream could potentially serve as an independent biomarker of inflammation in COVID-19 patients with T2DM.

## 1. Introduction

The coronavirus disease of 2019 (COVID-19) was determined to be triggered by the severe acute respiratory syndrome coronavirus 2 (SARS-CoV-2), which possessed the capacity to spread quickly throughout the majority of the world’s nations and also had a serious impact on the lives of billions of people [1,2]. As per the recent epidemiological data of the World Health Organization (WHO), around 774,824,115 positive confirmed COVID-19 cases have been reported worldwide [3]. SARS-CoV-2 is transferred mostly via droplets as well as close contact with symptomatically infected cases [4,5,6,7] and the common manifestations of the disease include respiratory failure, shock, kidney failure, cardiovascular damage, myocardial infarction, arrhythmias or liver failure [8,9]. SARS-CoV-2 infection may lead to the “cytokine storm”, also referred to as “cytokine release syndrome”, characterized by the sudden rise in circulating amounts of cytokines that are pro-inflammatory, including interleukin-6 (IL-6), IL-12, IL-17, IL-18, IL-33, tumor necrosis factor-alpha (TNF-α), and C-reactive protein (CRP) [10]. This cytokine storm likely inhibits both innate and adaptive immunity to the infection [11] and induces tissue disruption and an overall reduction in muscle protein synthesis in a combinatorial manner [12].

Type 2 diabetes mellitus (T2DM) is an illness defined by hyperglycemia, insulin resistance, and relative insulin insufficiency. Inflammation, endothelial dysfunction, and thrombosis are all prompted by worsening hyperglycemia, caused by the production of oxidative stress and further lead to abnormalities of glucose metabolism and hypercoagulability [13]. The link of diabetes with SARS-CoV-2 triggers cascading consequences, such as potentiating cytokine release and oxidative stress, which lead to end organ damage. Moreover, people with a COVID-19 infection are more likely to develop hyperglycemia as a result of it, which causes the angiotensin converting enzyme (ACE-2) receptors (promoters of SARS-CoV-2 entry) to be more glycosylated and promote viral multiplication [12,14]. 

Evidence also suggests that immune activation induces overexpression of insulin receptors by lymphocytes, further resulting in dysregulated insulin signaling in immune cells among individuals with COVID-19 and diabetes [15]. Cardiovascular system (CVS) risk factors can exacerbate COVID-19, causing chronic effects that explain cardiac pathologies to worsen and decompensate, followed by an acute onset of new cardiac complications, as demonstrated in 12% of patients who reported to be hospitalized due to SARS-CoV-2 infection [16].

Protease-activated receptors (PARs), belong to a subset of the G protein-coupled receptors, consisting of four members, namely PAR-1-4 [17]. PAR-1 is found in various cell types, particularly cardiomyocytes, neurons, epithelial cells, immune cells, and even platelets [18]. Previously referred to as the thrombin receptor, PAR-1 is involved in inflammation and coagulation. Its function in the hypercoagulable state and ischemia linked to SARS-CoV-2 has drawn a lot of attention. Endothelial cells and lung cells both express PAR-1. It is strongly expressed in the wounded lung’s extravascular and intravascular compartments, and it is linked to prothrombotic and endothelial dysfunction [17]. According to an experimental study, the activation of PAR-1 and PAR-2 in the respiratory epithelium leads to an overproduction of cytokines including IL-6 and IL-8, followed by the activation of the metalloproteinase MMP-12 and MMP-9, signaling pathways triggered by antigens, and ultimately, an increased sensitivity of the airways, known as airway hyper-reactivity [19]. A study by Fang shows that thrombin, when added to P815 mouse mast cells, caused PAR-1 expression, which in turn caused the release of TNF-α, the chemical messengers CCL2, CXCL1, and CXCL-5, the cytokines IL-2 and IL-6, and vascular endothelial growth factor [20]. Another study conducted on human beings and mice [21] demonstrated that lung tissue and epithelial cells, which are producers of thrombin, stimulated PAR-1 during liver infection and even resulted in fibrotic remodeling and the breakdown of the airway barrier [22]. Moreover, some findings suggest that mice with a genetic PAR-1 deficit or subjected to the PAR-1 antagonist SCH79797 were resistant to influenza A (H1N1 strain A/PR/8/34). Furthermore, after contracting HA type 1 and neuraminidase type 1/Puerto Rico strain 8/34 (H1N1/PR8), PAR-1 deficient mice exhibited lower levels of C-X-C motif chemokine ligand 10 (CXCL10) presence in the lung than the infected PAR-1 +/+ mice, which further correlated with higher levels of virus and inflammatory mediator expression [23].

Taken together, these discoveries highlight the involvement of PAR-1 in the innate reaction to influenza infection [23]. It is also known that PAR-1 is expressed in human platelets, and the activation of platelets has also been shown in COVID-19 patients [24]. Overall, the above findings are conclusive about the role of PAR-1 in inflammation during various disease states, such as viral infections and influenza. Therefore, the main aim of this study is to measure serum PAR-1 levels and deduce their association with inflammatory markers and clinical characteristics of T2DM affected COVID-19 patients. Both DM and COVID-19 are inflammatory conditions, with inflammation being a hallmark feature of COVID-19 infection. It is critical to comprehend how diabetes affects COVID-19 and vice versa, since doing so may help prognostic assessments and provide insights into how the disease progresses.

## 2. Results

### 2.1. Clinical Characteristics

Table 1 demonstrated baseline clinical and demographic characteristics of the 50 COVID-19 non-DM subjects (control group) and 50 COVID-19 T2DM subjects. No significant relation was found between the COVID-19 non-DM and COVID-19 T2DM subjects (*p* = 0.3655) in terms of age. COVID-19 DM subjects presented with high BMI (29.21 ± 3.52) when compared with control group (25.25 ± 4.01). In contrast to the control group, the COVID-19 T2DM groups exhibited a significant increase in PAR-1, NT-proBNP, HbA1c, IL-6, D-dimer, TNF-α, CRP, and homocysteine (Figure 1).

### 2.2. Comparison of DM Patients and Healthy Controls Subjects

Statistically significantly higher HbA1c was reflected in patients with COVID-19 T2DM when compared with the control group (Table 1). Levels of PAR-1 were significantly raised in subjects with COVID-19 T2DM versus the control group (50.76 ± 31.21 vs. controls 23.99 ± 12.68) (Table 1). Moreover, CRP levels in patients with COVID-19 T2DM significantly differed when compared with control subjects (57.13 ± 2.67 vs. controls 42.4 ± 4.15, *p* = 0.0029) (Table 1). Higher levels of D-dimer (*p* = 0.0306), NT-proBNP, IL-6, homocysteine, and TNF-α appeared in COVID-19 DM patients than in non-DM COVID-19 subjects, and these were found to be statistically significant (*p* < 0.0001) (Table 1).

### 2.3. Spearman Correlation of PAR-1 with Inflammatory Markers and Clinical Variables in COVID-19 DM Patients

Results revealed significant positive relationship between PAR-1 and BMI (*r* = 0.9989), HbA1c (*r* = 0.9987), D-dimer (*r* = 0.9994), NT-proBNP (*r* = 0.9994) CRP (*r* = 0.9995), IL-6 (*r* = 0.9994), homocysteine (*r* = 0.9993), and TNF-α (*r* = 0.9997) with *p* < 0.0001) (Figure 2 and Table 2).

## 3. Discussion

COVID-19 is an infectious disease prompted by the SARS-CoV-2 virus. The severity of COVID-19, a multi-organ illness, may vary from being asymptomatic to quite severe [25,26]. The respiratory system is the foremost organ system affected by COVID-19, and it may also have a major impact on the organs such as the heart, kidney, liver, muscles, skin, and neurological system [27].

PAR are G-protein-coupled receptors, expressed by a variety of different cell types. For the potential therapy of a number of illnesses, including thrombosis, atherosclerosis, restenosis inflammatory processes, cancer metastasis, and stroke, PAR-1 is a promising drug development target [28]. Numerous pieces of evidence point to a connection between the activation of PAR-1 and the development and release of pro-inflammatory cytokines [17,29,30]. According to this study, blocking PAR-1 may prevent inflammatory mediators from being released when a person has diabetes [30]. Potential treatment strategies for regulating hyperactivated megakaryocytes (MKs) have been investigated, including the expression of PAR-1 in 138 patients. The PAR-1 levels in both S100A8/A9+ MK populations rose both considerably and similarly. When taken as a whole, these findings imply that MKs infected with viruses have an NF-κB-mediated inflammatory signature, are hyperactivated, and may be anemic [31]. Another study of PAR-1 in DM nephropathy shows that after the establishment of diabetes, PAR-1-deficient animals showed decreased kidney damage, as seen by reduced proteinuria, plasma cystatin C levels, mesangial area expansion, and tubular atrophy. The matrix proteins fibronectin and collagen IV were expressed and proliferated more readily in mesangial cells in vitro as a result of PAR-1 signaling. On the other hand, in DM PAR-1-deficient animals, a decrease in both proliferation and fibronectin deposition was seen [32].

In a study of 1996 patients who had at least one BMI measurement taken both before and after the COVID-19 epidemic, patients with a higher BMI were discovered to be at higher risk [33]. Similar results were obtained in our study, where BMI levels were highly elevated in T2DM affected COVID-19 subjects, and also exhibited a strong and positive correlation with serum PAR-1 levels.

In a study involving 200 T2DM subjects with COVID-19 infection, the expressions of CRP were highly increased compared to non-DM patients with COVID-19 [34]. CRP, as a sensitive biomarker of inflammation, holds potential as an early predictor in patients affected by COVID-19. In addition, Vietnamese research showed that the affected COVID-19 individuals (regardless of the disease’s stage) presented with increased CRP levels [32]. Furthermore, research evidence suggests that higher levels of CRP are positively associated with worsening COVID-19 pneumonia [35,36]. Parallel investigation results were also determined in present research, where CRP were increased in T2DM-affected COVID-19 subjects, and also depicted a strong and positive correlation with serum PAR-1 concentrations.

Evidence suggests enhanced serum IL-6 levels in severe COVID-19 subjects compared to mild-moderate COVID-19 patients [37]. Another study involved 91 patients with COVID-19 infection, out of whom 14 (19.72%) were diagnosed with T2DM and exhibited elevated levels of IL-6 compared to non-DM patients with COVID-19 [38]. Our findings also suggested an elevation in IL-6 levels in T2DM-affected COVID-19 subjects, which further showed a strong and positive correlation with serum PAR-1 levels.

A serum cytokine analysis, involving 207 COVID-19 patients, presented an elevation in levels of TNF-α in those with severe COVID-19 [39]. In our current study, TNF-α levels were also determined to be elevated in T2DM-affected COVID volunteers, which also showed a positive correlation with PAR-1 levels.

Furthermore, a study by Gao and colleagues on patients with severe COVID-19 infection, who were analyzed for NT-proBNP, showed a high mortality rate in COVID-19 patients who exhibited high levels of the biomarker. Moreover, NT-proBNP also had a correlation with other inflammatory markers [40]. Similar findings were reported in our study, where NT-proBNP was also found to be elevated in the DM COVID-19 subjects, followed by a strong and positive correlation with serum PAR-1 levels. One of the experimental studies involving 117 COVID-19 patients and 34 non-COVID-19 patients showed high levels of homocysteine and D-dimer in COVID-19 patients, which further provided information about the severity of the condition in them [41]. These findings were consistent with our study, where high levels of homocysteine and D-dimer were also observed in COVID-19 DM subjects, along with a positive correlation with serum PAR-1 levels.

Collectively, our study revealed that COVID-19-affected T2DM patients exhibited higher levels of several inflammatory mediators, including CRP, IL-6, and TNF-α in comparison to COVID-19 non-DM patients. So, elevated PAR-1 levels could serve as a prognostic marker for COVID-19 severity and outcomes in T2DM patients. Studies have shown that DM and COVID-19 severity is often associated with an exacerbated inflammatory response [12,42].

PAR-1, being involved in inflammation regulation, may reflect the severity of the inflammatory response and consequently predict the progression of the disease and the likelihood of complications. The presence of elevated PAR-1 levels sheds light on the underlying pathophysiological mechanisms at play in COVID-19 patients with T2DM. PAR-1 is known to be involved in various inflammatory processes, including endothelial dysfunction, platelet activation, and cytokine release [17]. Its increased serum levels suggest dysregulation of these processes, potentially contributing to the increased inflammation observed in the COVID-19 and diabetic patients. A better understanding of the role of PAR-1 in the inflammatory response in COVID-19 patients with T2DM, and drugs targeting PAR-1 or its downstream signaling pathways, could be potential novel therapeutic targets to mitigate inflammation and improve outcomes in these patients.

The current study also has some limitations. To identify a causal relationship between PAR-1 and other inflammatory biomarkers, genetic and multicenter studies on large sample sizes are necessary to further specify and validate the findings of the current study and to confirm the prediction accuracy of PAR-1 in the study population. It is imperative to further establish the physiological and pathological relationship between PAR-1 and other inflammatory diseases.

## 4. Materials and Methods

### 4.1. Study Design and Patients

In this cross-sectional study, the patients, aged between 30 and 60 years, were recruited from the COVID-19 isolation area of Tertiary Care Hospital, India, in 2023, after approval by Institutional Human Ethical Committee (IHEC) (EC/NEW/INST/2023/531/180). COVID-19 was confirmed in patients by using the reverse transcription-polymerase chain reaction (RT-PCR) technique, and these individuals underwent screening for both primary and secondary infections, respectively. We recruited only T2DM patients with COVID-19 and non-DM patients with COVID-19. Inclusion criteria for the diabetic COVID-19 patients followed the American Diabetes Association guidelines which say that the HbA1c level for the diabetic patients should be above 6.5%. The exclusion criteria comprised children, pregnant females, cancer individuals, respiratory disorders (except COVID-19), and comorbidities (except diabetes). Before including a participant in the research, informed consent forms were collected from each patient once they had been made aware of the study. The research adhered to the principles outlined in the Declaration of Helsinki and followed by ICH E6GCP standards, with the necessary modifications.

### 4.2. Biochemical Assessment

All the study participants gave 10 mL of their venous blood samples for the study. Centrifugation was used to separate the sera from the entire blood, and it was performed for 15 min at 1300 rpm. HbA1c, TNF, IL-6, IL-8, and CRP ELISA kits were obtained from Krish-gen Bio-systems in Mumbai, India, and PAR-1 kit were obtained from Wuhan, China. The PAR-1 Fine Test ELISA kit had an analytical sensitivity lower than 4.95 g/mL. All the laboratory procedures were carried out as per the instructions provided in the kit. All the assays were conducted as per standard principles. In contrast to the intra-assay rates of variation, which were all less than 10%, the inter-assay rates of variation for PAR-1, HbA1c, TNF-α, IL-6, IL-8, and CRP were found to be less than 8%, 12%, 12%, and 12%, respectively.

### 4.3. Statistical Analysis

The arithmetic mean and standard deviation (SD) were run to depict all the non-parametric and normally distributed variables. Using unpaired Student’s t-tests, significance between the study groups was represented. Spearman’s correlation evaluated the univariate analysis, and the values were deemed statistically significant if the *p*-value was less than 0.05. The statistical software Graph Pad Prism 7.04 was used to perform all the data analysis.

## 5. Conclusions and Future Directions

This study provided the first evidence that PAR-1 levels rise in COVID-19-affected T2DM patients, and it identified a positive correlation between serum BMI, PAR-1 levels, CRP, HbA1c, D-dimer, homocysteine, NT-proBNP, and other inflammatory mediators. According to this study, increased PAR-1 levels in serum could serve as a valuable diagnostic indicator for inflammation in COVID-19-affected T2DM patients. However, the concluded findings must be confirmed in a large number of T2DM patients with COVID-19.

The molecular, biochemical, and structural aspects of PAR-1 physiology, and mechanism of action still need to be investigated. Further research is required to completely understand the signaling mechanisms through which PAR-1 acts. If these pathological mechanisms and the subsequent signaling processes are identified, it will be easier to use PAR-1 biology therapeutically to treat and manage inflammatory illnesses. Furthermore, clinical research is continuously evolving in terms of diagnostic tools/biomarkers and therapeutic targets. Biomarker discovery could facilitate diagnostics and personalized treatment strategies. Thus, the present research provides a potential application of PAR-1 to assess dysfunctionalities caused by inflammation and translate these findings into clinical applications for improved management of inflammatory diseases. Additionally, research on PAR-1 may explore therapeutic applications, such as developing PAR-1 inhibitors for targeted treatment in inflammatory diseases.

## Figures and Tables

**Figure 1 pharmaceuticals-17-00454-f001:**
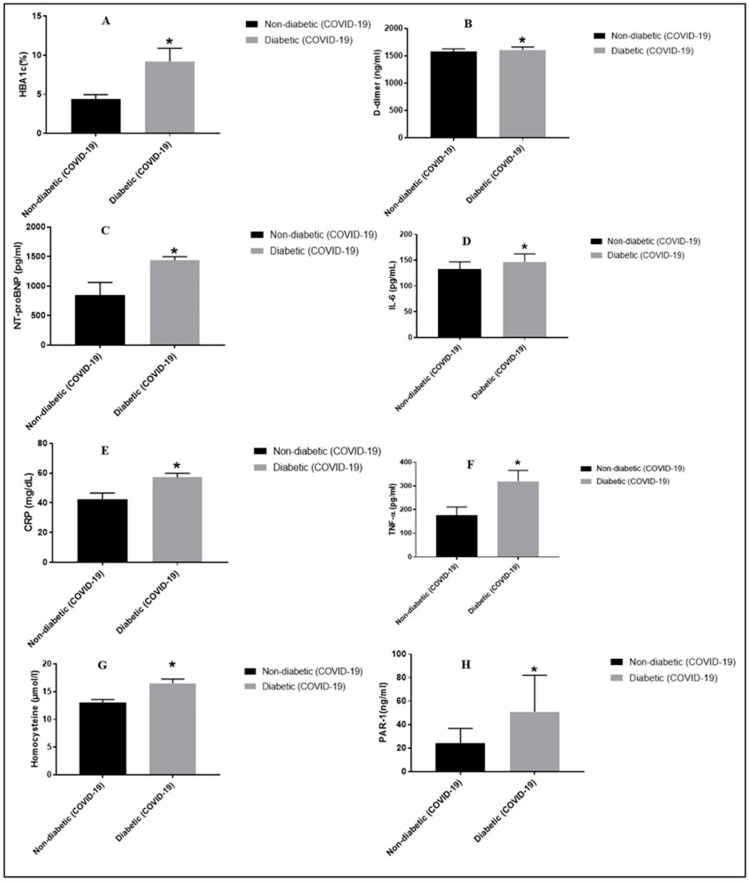
Interpretation of demographic, biochemical, cardiac, and inflammatory biomarker levels in COVID-19-affected DM and non-DM subjects. * *p* < 0.05 vs. the non-DM group, respectively. Panel (**A**) depicts HbA1c levels in T2DM vs. non-DM group; panel (**B**) depicts D-dimer levels in T2DM vs. non-DM group; panel (**C**) depicts NT-proBNP levels in T2DM vs. non-DM group; panel (**D**) depicts IL-6 levels in T2DM vs. non-DM group; panel (**E**) depicts CRP levels in T2DM vs. non-DM group; panel (**F**) depicts TNF-α levels in T2DM vs. non-DM group; panel (**G**) depicts homocysteine levels in T2DM vs. non-DM group; panel (**H**) depicts serum PAR-1 levels in T2DM vs. non-DM group.

**Figure 2 pharmaceuticals-17-00454-f002:**
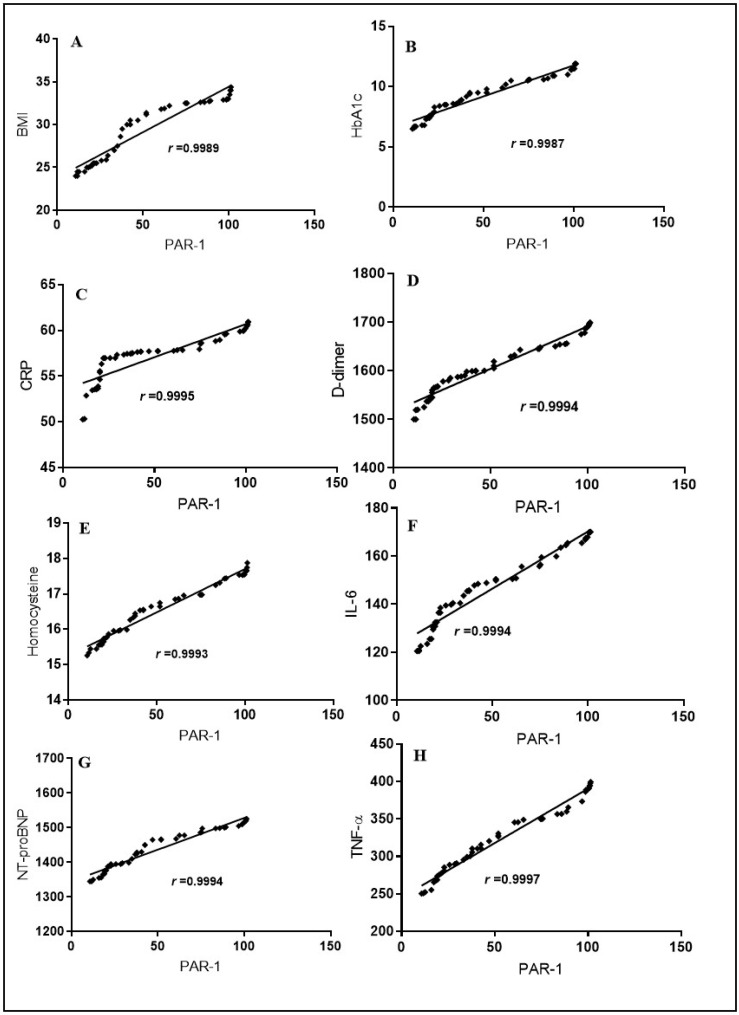
Spearman correlation among serum PAR-1 levels, BMI, HbA1c, the cardiac marker NT-proBNP, and other inflammatory markers in COVID-19-affected T2DM patients. The correlation is depicted between PAR-1 and BMI values in panel (**A**), PAR-1 and HbA1c levels in panel (**B**), PAR-1 and CRP levels in panel (**C**), PAR-1 and D-dimer levels in panel (**D**), PAR-1 and homocysteine levels in panel (**E**), PAR-1 and IL-6 levels in panel (**F**), PAR-1 and NT-proBNP levels in panel (**G**), and PAR-1 and TNF-α levels in panel (**H**).

**Table 1 pharmaceuticals-17-00454-t001:** Demographic and biochemical parameters of study groups (COVID-19 affected individuals with and without T2DM). Data expressed as mean ± SD. *p* < 0.05 was considered as statistically significant.

Variables	T2DM Patients with COVID-19	Non-DM Patients with COVID-19	*p* Value
n	50	50	-
Age (in years)	41.38 ± 7.58	39.83 ± 7.23	0.3655
Gender (Men/Women)	29/21	26/24	-
BMI (kg/m^2^)	29.21 ± 3.52	21.30 ± 2.11	<0.0001
HbA1c (%)	9.23 ± 1.66	4.39 ± 0.57	<0.0001
D-dimer (ng/mL)	1605.08 ± 56.57	1578.66 ± 48.72	0.0306
NT-proBNP (pg/mL)	1438.18 ± 59.48	852.82 ± 207.26	<0.0001
IL-6 (pg/mL)	146.84 ± 15.43	133.03 ± 13.67	<0.0001
CRP (mg/dL)	57.13 ± 2.67	42.4 ± 4.15	<0.0001
TNF-α (pg/mL)	319.09 ± 46.28	175.29 ± 35.10	<0.0001
Homocysteine (µmol/L)	16.50 ± 0.78	13.02 ± 0.57	<0.0001
PAR-1 (ng/mL)	50.76 ± 31.21	23.99 ± 12.68	<0.0001

**Table 2 pharmaceuticals-17-00454-t002:** The univariate analysis between serum PAR-1 levels and demographic, biochemical, and inflammatory parameters of COVID-19 affected T2DM subjects.

Variables	Univariate Analysis	
	*r*	*p* Value
PAR-1		
Age (years)	0.4712	0.0006
BMI (kg/m^2^)	0.9989	<0.0001
HbA1c (%)	0.9987	<0.0001
D-dimer (ng/mL)	0.9994	<0.0001
NT-proBNP (pg/mL)	0.9994	<0.0001
CRP (mg/dL)	0.9995	<0.0001
IL-6 (pg/mL)	0.9994	<0.0001
TNF-α (pg/mL)	0.9997	<0.0001
Homocysteine (µmol/L)	0.9993	<0.0001

HbA1c, Glycated hemoglobin; NT-proBNP, N-terminal pro-B-type natriuretic peptide; BMI, Body Mass Index; IL-6, Interleukin-6; CRP, C-reactive protein; PAR-1, protease-activated receptor; and TNF-α, tumor necrosis factor-α.

## Data Availability

The data and materials are available from the corresponding author upon request.

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
