# Peer review of "Unveiling the Role of PAR 1: A Crucial Link with Inflammation in Diabetic Subjects with COVID-19"

_pharmaceuticals, 2024, doi:10.3390/ph17040454_

Round 1

Reviewer 1 Report

Comments and Suggestions for Authors

In the paper entitled “Unveiling the Role of PAR 1: A Crucial Link with Inflammation in Diabetic Subjects with COVID-19” Ravinder Singh and collaborators examine the serum levels of PAR1, evaluating their association with different biochemical parameters and inflammatory biomarkers, in type 2 diabetes mellitus- affected COVID-19 patients, founding them as an independent biomarker of inflammation in COVID-19 patients with diabetes.

Given the known role of PAR-1 in inflammation during various disease states, the reported results are not surprising. 

I have some comments, reported below

First of all, in my opinion, the abstract should be remodeled and some background information should be inserted.

In the introduction section, I think there are typing errors in correspondence of lane 39 and lane 40, where are indicated the numbers of COVID-19 cases, check it.

It’s not completely clear to me (perhaps it would be enough to explain better in the text), why the authors decide to investigate the role of PAR-1 in COVID-19 patients with diabetes: what about the comparison between COVID-19 and NON COVID-19 patients? For the same reasons, have the authors evaluated NON COVID subjects suffering from diabetes? Considering that both the pathological conditions are linked to inflammation, where PAR-1 it’s known to play a role, do you expect additive effects when COVID-19 and diabetes co-exist? Can PAR-1 be considered a biomarker of inflammation in COVID-19 patients independently from the co-existence of diabetes condition (COVID-19 versus NON COVID-19 patients)?

Author Response

While thanking profusely the scholarly reviewer, we corrected all the errors which inadvertently crept into our manuscript. The authors are thankful to esteemed reviewers and editors for giving their valuable insights and suggestions which have made the present manuscript more appropriate and appealable. We have revised our manuscript according to the suggested guidelines. All the changes are highlighted in yellow.

Reviewer 1 Comments:

Comment 1: First of all, in my opinion, the abstract should be remodeled and some background information should be inserted.

Reply: We are thankful to the reviewers for helping us to improve our manuscript. As per the instructions, abstract has been revised and highlighted in the revised manuscript.

Comment 2: In the introduction section, I think there are typing errors in correspondence of lane 39 and lane 40, where are indicated the numbers of COVID-19 cases, check it.

Reply: We apologies for the typographical mistakes. As per the advice of respected learned reviewers, the manuscript has been revised and typographical mistakes have been corrected. -

Comment 3: It’s not completely clear to me (perhaps it would be enough to explain better in the text), why the authors decide to investigate the role of PAR-1 in COVID-19 patients with diabetes: what about the comparison between COVID-19 and NON COVID-19 patients? For the same reasons, have the authors evaluated NON COVID subjects suffering from diabetes? Considering that both the pathological conditions are linked to inflammation, where PAR-1 it’s known to play a role, do you expect additive effects when COVID-19 and diabetes co-exist? Can PAR-1 be considered a biomarker of inflammation in COVID-19 patients independently from the co-existence of diabetes condition (COVID-19 versus NON COVID-19 patients).

Reply: Both Diabetes Mellitus and COVID-19 are inflammatory conditions, with inflammation being a hallmark feature of COVID-19 infection. Research literature suggests that PAR-1 linked to inflammation, may be a potential and emerging biomarker in inflammatory conditions (Sibia, 2023). Due to increased inflammation, the convergence of COVID-19 and diabetes may make the patient's condition worse and raise the risk of complications such as secondary infections and cardiovascular diseases. As a result, research has been conducted to examine how diabetes and COVID-19 interact, taking into account the possibility that this could worsen patient outcomes. In fact, there are serious hazards associated with COVID-19 and diabetes co-occurring. It is critical to comprehend how diabetes affects COVID-19 and vice versa since doing so may help prognostic assessments and provide insights into how the disease progresses.

Sibia, R. S., Sood, A., Subedi, A., Sharma, A., Mittal, A., Singh, G., ... & Goyal, S. (2023). Elevated serum PAR‐1 levels as an emerging biomarker of inflammation to predict the dengue infection severity. Journal of Medical Virology95(1), e28152.].

Reviewer 2 Report

Comments and Suggestions for Authors

The introduction lacks specific information on the role of PAR-1 in inflammation in the context of COVID-19. Please provide a more comprehensive background on the significance of PAR-1 and its potential implications for diabetic patients with COVID-19. Update the introduction with the latest epidemiological data on COVID-19.

Provide a more detailed description of the study design and patient recruitment, including the inclusion and exclusion criteria. Include information on ethical considerations and approval obtained for the study. Provide a more detailed explanation of the ELISA assay protocol used to measure serum PAR-1 levels.

Include appropriate statistical tests and provide a detailed interpretation of the findings. Expand on the correlation between serum PAR-1 levels and other parameters, including their strength and significance. Present the data in tables or figures for better clarity.

Provide a comprehensive analysis and interpretation of the results in the context of existing literature.

Expand on the implications of increased serum PAR-1 levels as an independent biomarker of inflammation in COVID-19 patients with T2DM.

Discuss the limitations of the study and suggest areas for future research.

Major Revisions Required

Please address the major revisions suggested above to improve the clarity, scientific rigor, and overall quality of the manuscript. Once the revisions are made, the paper can be re-evaluated for further consideration.

Please note that failure to address the major revisions adequately may result in the rejection of the manuscript.

Author Response

REBUTTAL LETTER

While thanking profusely the scholarly reviewer, we corrected all the errors which inadvertently crept into our manuscript. The authors are thankful to esteemed reviewers and editors for giving their valuable insights and suggestions which have made the present manuscript more appropriate and appealable. We have revised our manuscript according to the suggested guidelines. All the changes are highlighted in yellow.

Reviewer 2 Comments:

Comment 1: The introduction lacks specific information on the role of PAR-1 in inflammation in the context of COVID-19. Please provide a more comprehensive background on the significance of PAR-1 and its potential implications for diabetic patients with COVID-19. Update the introduction with the latest epidemiological data on COVID-19.

Reply:  As per suggestions, introduction section has been updated and highlighted in the revised manuscript.

Comment 2: Provide a more detailed description of the study design and patient recruitment, including the inclusion and exclusion criteria. Include information on ethical considerations and approval obtained for the study. Provide a more detailed explanation of the ELISA assay protocol used to measure serum PAR-1 levels.

Reply: As per the directions given by the esteemed reviewer, methodology section has been updated and highlighted in the revised manuscript. Detailed procedure of PAR-1 serum estimation by the ELISA assay protocol has been explained below-

Principle:

This kit was based on sandwich enzyme-linked immune-sorbent assay technology. The capture antibody was pre-coated onto 96-well plates and biotin conjugated antibody were used as detection antibodies. The standards test samples and biotin conjugated detection antibody were added to the wells subsequently, washed with wash buffer. HRP-Streptavidin was added and unbound conjugates were washed away with wash buffer. TMB substrates were used to visualise HRP (horseradish peroxidase) enzymatic reaction. TMB was catalysed by HRP to produce a blue colour product that changed into yellow after adding acidic stop solution. The density of yellow was proportional to the target and was calculated.

Material Required:

  1. Microplate reader
  2. Automated plate washer
  3. 37oC incubator
  4. Precision single and multi-channel pipette and disposable tips
  5. Clean tubes and Eppendorf tubes
  6. Deionized or distilled water.

Sample Collection:

Blood sample was placed at room temperature for 2 hours and centrifugation was done for 20 minutes at approximately 1000xg. The supernatant was collected and the assay was carried out immediately. Blood collection tubes were disposable, non-pyrogenic and non-endotoxin.

Sample Dilution:

We estimated the concentration of target protein in the test sample, and a proper dilution factor was selected to make the diluted target protein concentration fall into the optimal detection range of the kit. The sample was diluted with the provided dilution buffer. The test sample was mixed well with the dilution buffer. And also, standard curves and sample were made.

Reagent Preparation:

All reagents and samples were brought to room temperature for 20 minutes before use.

  1. Wash Buffer

Crystals formed in the concentrate were warmed with 40°C water bath (Heating temperature not exceeded 50°C) and were mixed gently until the crystals were completely dissolved. The solution was cooled to room temperature before use.

30ml (15ml for 48T) Concentrated Wash Buffer was diluted into 750ml (375ml for 48T) Wash Buffer with distilled water.

  1. Standards

1)         1 ml Sample Dilution Buffer was added into one Standard tube (labelled as zero tube), kept at room temperature for 10 minutes and mixed thoroughly.

2)         7 EP tubes were labelled with 1/2, 1/4, 1/8, 1/16, 1/32, 1/64 and blank respectively. 0.3ml of the Sample Dilution Buffer was added into each tube. 0.3ml of the above Standard solution (from zero tube) was added into 1st tube and were mixed thoroughly. 0.3ml from 1st tube was transferred to 2nd tube and mixed thoroughly. 0.3ml from 2nd tube was transferred to 3rd tube and mixed, and so on the sample dilution buffer was used for the blank control.

  1. Preparation of Biotin-Labelled Antibody Working solution:

       It was prepared within 1 hour before experiment.

  • The required volume of the working solution was calculated: (0.1 ml/well *Quality of wells).
  • Biotin-detection antibody was diluted with antibody dilution buffer at 1:100 and mixed thoroughly.
  1. Preparation of HRP-Streptavidin Conjugate (SABC) working Solution:

It was prepared within 30 minutes before experiment.

  • The required volume of the working solution was calculated (0.1 ml/well *Quantity of wells).
  • The SABC was diluted with SABC Dilution Buffer at 1:100 and mixed thoroughly.

Assay Procedure:

Samples and reagents were diluted and mixed completely and evenly. Before adding TMB into wells, TMB Substrate was equilibrated for 30 minutes at 37°C. The standard curve for each test was plotted.

  • Standard, test samples (diluted at least 1/2 with Sample Dilution Buffer), control (blank) wells on the pre-coated plate were set respectively, and then, their positions were recorded. Each standard and sample in duplicate were measured. The plate was washed 2 times before adding standard, sample and control (blank) wells.
  • Prepared Standards: 100µl of zero tube, 1 tube, 2tube, 3"tube, 4"tube, 5"tube, 6" tube and Sample Dilution Buffer (blank) were aliquoted into the standard wells.
  • Samples were added: 100µl of properly diluted sample was added into test sample wells.
  • Incubated: The plate was sealed with a cover and incubated at 37°C for 90 minutes.
  • Washed: The cover was removed and the plate content was discarded, and the plate was washed 2 times with Wash Buffer. The wells were not allowed to dry completely at any time.
  • Biotin-labelled Antibody: 100µl Biotin-labelled antibody working solution was added into above wells (standard, test sample and blank wells). The solution was added at the bottom of each well without touching the sidewall, the plate was covered and incubated at 37°C for 60 minutes.
  • Washed: The cover was removed, and the plate was washed 3 times with Wash Buffer, and Wash Buffer was allowed to stay in the wells for 1-2 minutes each time.
  • HRP-Streptavidin Conjugate (SABC): 100µl of SABC Working Solution was added into each well, the plate was covered and incubated at 37°C for 30 minutes.
  • Washed: The cover was removed and the plate was washed 5 times with the Wash Buffer, and the wash buffer was allowed to stay in the wells for 1-2 minutes each time.
  • TMB Substrate: 90µl TMB Substrate was added into each well, the plate was covered and incubated at 37°C in dark within 10-20 minutes. (Note: The reaction time can be shortened or extended according to the actual colour change, but not more than 30 minutes). The reaction was terminated when the apparent gradient appeared in standard wells.)
  • Stopped: 50µl Stop Solution was added into each well. The colour turned to yellow immediately. The adding order of Stop Solution was same as the TMB Substrate Solution. OD Measurement: The O.D. absorbance at 450nm was read in Microplate Reader immediately after adding the stop solutions.

Comment 3: Include appropriate statistical tests and provide a detailed interpretation of the findings. Expand on the correlation between serum PAR-1 levels and other parameters, including their strength and significance. Present the data in tables or figures for better clarity.

Reply: In response to mentioned comment, comparison between the groups has been examined by unpaired student t-test and Spearman correlation and further univariate regression analysis between serum PAR1 levels and other parameters and these statistical tools presents an authenticated data as per methodology of the study. The same has been updated in Tables in manuscript as suggested.

Comment 4: Provide a comprehensive analysis and interpretation of the results in the context of existing literature.

 Reply: As per the comments given by the respected reviewer, discussion part has been updated and highlighted in the revised manuscript.

Comment 5: Expand on the implications of increased serum PAR-1 levels as an independent biomarker of inflammation in COVID-19 patients with T2DM.

Reply: As per the suggestions, separate paragraph on the implications of increased serum PAR-1 levels as an independent biomarker of inflammation in COVID-19 patients with T2DM has been added in discussion section and highlighted in the revised manuscript.

Comment 6: Discuss the limitations of the study and suggest areas for future research.

Reply: As per the instructions, limitations of the study and future directions has been introduced and highlighted in the separate paragraph in the revised manuscript.

Reviewer 3 Report

Comments and Suggestions for Authors

Singh et al. present a manuscript on the role of PAR1 on COVID-19 in diabetic cohort. The manuscript is nicely written and there is only one major comment this reviewer has. Have the authors validated the PAR1 assay in their hands? It seems that the reported ranges are quite high (both for diabetic and non-diabetics). Can authors recruit healthy age-matched controls and include in the manuscript? 

Comments on the Quality of English Language

No editing required. 

Author Response

Not Applicable

Round 2

Reviewer 1 Report

Comments and Suggestions for Authors

In the paper entitled “Unveiling the Role of PAR 1: A Crucial Link with Inflammation in Diabetic Subjects with COVID-19” Ravinder Singh and collaborators examine the serum levels of PAR1, evaluating their association with different biochemical parameters and inflammatory biomarkers, in type 2 diabetes mellitus- affected COVID-19 patients, founding them as an independent biomarker of inflammation in COVID-19 patients with diabetes.

In the revised version the authors have clarified the doubts of the reviewer and adequately improved the manuscript as required

Author Response

/

Reviewer 2 Report

Comments and Suggestions for Authors

Accepted 

Author Response

/

Reviewer 3 Report

Comments and Suggestions for Authors

The authors addressed the main question this reviewer had. 

Author Response

/